# Multilingual capabilities of GPT: A study of structural ambiguity

**Myung Hye Yoo[1], Joungmin Kim[2], Sanghoun Song[3]***

1 Department of English Language and Literature, Chungnam National University, Daejeon, South Korea, 2 Department of Japanese Language and Literature, Korea University, Seoul, South Korea, 3 Department of Linguistics, Korea University, Seoul, South Korea

* sanghoun@korea.ac.kr

## Abstract

This study examines the multilingual capabilities of GPT, focusing on its handling of syntactic ambiguity across English, Korean, and Japanese. We investigate whether GPT can capture language-specific attachment preferences or if it relies primarily on English-centric training patterns. Using ambiguous relative clauses as a testing ground, we assess GPT's interpretation tendencies across language contexts. Our findings reveal that, while GPT (GPT-3.5-turbo, GPT-4-turbo, GPT 4o)'s performance aligns with native English speakers' preferred interpretations, it overgeneralizes this interpretation in Korean and lacks clear preferences in Japanese, despite distinct attachment biases among native speakers of these languages. The newer, smaller-scale models—o1-mini and o3-mini—further reinforce this trend by closely mirroring English attachment patterns in both Korean and Japanese. Overall results suggest that GPT's multilingual proficiency is limited, likely reflecting a bias toward high-resource languages like English, although differences in model size and tuning strategies may partially mitigate the extent of English-centric generalization. While GPT models demonstrate aspects of human-like language processing, our findings underscore the need for further refinement to achieve a more nuanced engagement with linguistic diversity across languages.

## Introduction

In recent years, large language models (LLMs) have demonstrated remarkable abilities across a range of natural language processing tasks, including chain-of-thought reasoning [1–3], in-context learning [4,5], word-in-context semantic judgment [6], and even professional exams [7–9]. In particular, GPT models, developed by OpenAI, play a dominant role in both academic research and real-world AI applications, powering tools like ChatGPT and numerous commercial and educational platforms. This widespread adoption positions GPT as a compelling and practical foundation for investigating how LLMs handle complex tasks.

**Data availability statement:** All relevant data are within the manuscript and its Supporting Information files.

**Funding:** This work was supported by the NRF under the project BK21 FOUR (4299990414427). There was no additional external funding received for this study.

**Competing interests:** NO authors have competing interests.

ChatGPT leverages supervised learning through human-annotated examples to produce targeted outputs for given prompts. Additionally, human rankings of generated outputs are used to develop a reward model that optimizes ChatGPT's performance through reinforcement learning. As one of the leading LLMs, ChatGPT rapidly captured widespread attention after its release on November 30, 2022, amassing 100 million users within its first two months [10]. It has gained significant attention for its coherent responses across diverse tasks, including creative writing [11], image generation [12], computer programming or coding [13,14], sentiment analysis [15,16], and various annotation tasks [17–19].

Further reports highlight GPT models' success in handling complex, domain-specific tasks, in various domains such as medicine [7,8,20,21], law [22–24], public health [25], economics [26], and programming [27,28]. This has sparked enthusiasm across fields for potential applications of ChatGPT as professional assistants [29–32]. Such findings suggest that LLMs may significantly impact various industries and applications in the near future.

With the global spread of LLM-based services like ChatGPT, their use has extended to non-English-speaking communities. As ChatGPT has become widely adopted in English-speaking contexts across multiple domains, assessing its performance in specialized fields for non-English languages is increasingly essential. However, most tasks are predominantly focused on English, and much of ChatGPT's performance has been assessed primarily in the English language. Hence, even though ChatGPT is trained on extensive, large-scale datasets from diverse sources, it often shows a proficiency bias toward high-resource languages [33]. This creates the possibility of issues such as replication of information from high-resource languages in outputs without proper citation or application of English-trained knowledge to non-English contexts. This raises questions about whether multilingual LLMs are truly versatile across languages—specifically, whether GPT models can be effectively used in other languages or if more language-specific technologies are needed. To investigate GPT's multilingual capabilities, recent studies have begun assessing its performance across various tasks in non-English languages. These comprehensive evaluations cover areas such as reasoning abilities, language identification, and machine translation with diverse languages [34,35], as well as specific applications like medical licensing exams in Japanese [36] and Chinese [37] and tasks like grammatical error correction or cross-lingual summarization in languages such as Chinese and German [38,39]. Overall, studies indicate that GPT performs reasonably well in non-English languages, albeit with some inaccuracies. For instance, while its grammatical error correction is generally effective at the sentence level, it falls short at the discourse level in languages like German and Chinese [38]. Bang et al. [34], conducting a series of multilingual, multimodal, multitasks on reasoning, hallucination, and interactivity, found GPT unreliable. Furthermore, [35,40] showed that GPT tends to perform better with English prompts, even when tasks and inputs are designed for other languages, demonstrating its English-language bias.

While ChatGPT has been evaluated for many different domains, relatively few studies examine its multilingual proficiency in linguistics. Our study addresses this

gap by evaluating GPT's multilingual ability within the linguistic domain: resolution of ambiguities. Like other fields, LLMs exhibit behavior resembling human linguistic processing [41–43]. Ambridge and Blything [44] even have demonstrated that LLMs outperforms traditional theoretical linguists in predicting human judgments in certain type of structure. Meanwhile, much of the research has been conducted primarily on English [45].

To offer deeper insights into multilingual applications of GPT, we focus specifically on its interpretation of ambiguous sentences across languages. As GPT's capabilities evolve, it is expected to prioritize multiple valid interpretations in a human-like manner [46]. Effectively managing ambiguity is a crucial aspect of human language comprehension, allowing speakers to anticipate misunderstandings and listeners to adjust interpretations for smooth communication. For language models like GPT, mastering this skill is critical—not only for its effectiveness in conversational interfaces and writing aids but also for achieving communication skills that resemble human interaction. A particularly insightful way to assess GPT's multilinguality and conversational sophistication is by examining its handling of ambiguous sentences, where human preferences for resolution vary significantly across languages. For instance, Japanese and Korean speakers often show opposite tendencies to English speakers. Our study focuses on investigating GPT's ability to reflect language-specific interpretations in English, Korean, and Japanese while resolving structural ambiguities. We assess whether it can apply unique linguistic knowledge or if it primarily relies on English-based training data. By examining GPT's handling of ambiguous sentences across these languages, we aim to gain crucial insights into its capacity for genuinely multilingual, human-like communication. In addition, we compare multiple versions of GPT—from GPT-3.5-urbo to the recent o1-mini and o3-mini models—to examine how variations in tuning techniques, model scale, and optimization goals influence interpretive behavior across languages. Such comparisons allow us to trace the developmental trajectory of GPT models' multilingual capabilities and to identify key factors underlying language-specific parsing behavior. In this context, we aim to gain insight not only into the practical implications of GPT model selection but also into systematic strategies for enhancing linguistic adaptability across diverse languages and language families.

## Previous studies on attachment ambiguities

### Relative clause attachment ambiguity in psycholinguistics

Sentences are not always clearly written and may remain ambiguous even after they are terminated. For example, sentences containing complex noun phrases that are modified with a relative clause, known as attachment ambiguity of relative clauses, are typically perceived as ambiguous because relative clauses can modify either the first noun phrase (NP) or the second NP.

| (1) | Someone shot the servant[HIGH] of the actress[LOW] who was on the balcony. |
|-----|-----|
|     | (Cuetos & Mitchell [47], 1988) |

In the above sentence (1), it is not clear who exactly was on the balcony- *the servant* or *the actress*. This is because for the given complex NP clause (i.e., *the servant of the actress who was on the balcony*), the relative clause is grammatically correct regardless of whether it is attached to either the first NP (NP1, *the servant*) or the second NP (NP2, *the actress*). In English structure, NP1 is located structurally higher than NP2. We call "High attachment (HA)" to refer to the attachment to the structurally higher NP, that is, NP1 in English, whereas "Low attachment (LA)" refers to the attachment to the structurally lower NP, that is, NP2 in English.

Although the structural ambiguity associated with the sentence (1) is not resolved and has no correct answers, readers often have a tendency to select one NP over the other, with the preference tendency differing across various languages. LA is preferred by speakers of English [48–51], Chinese [52], Romanian [53], and Arabic [54], favoring to parsing the clauses to low NPs (i.e., *the actress*). HA is preferred by speakers of Spanish [47–49,55], Brazilian and European

Portuguese [56–58], Korean [59–63], Japanese [64], Dutch [65,66], French [67,68], German [69], and Greek [70], preferring to parse the clauses to high NPs (i.e., *the servant*).

As noted, while English native speakers prefer LA interpretation, Korean and Japanese native speakers have demonstrated HA preferences.

| (2) | | 'Someone shot the servant of the actress that was on the balcony.' | | | | | |
|---|---|---|---|---|---|---|---|
| | a. | Korean: | | | | | |
| | | Nwukwun-ka | palkhoni-ey | iss-nun | yepaywu-uy | kacengpwu-lul | sswassta |
| | | Someone-NOM | balcony-LOC | is-ADN | actress$_{[LOW]}$-GEN | servant$_{[HIGH]}$-ACC | shot |
| | b. | Japanese: | | | | | |
| | | Dareka-ga | barukonii-ni | iru | joyuu-no | mesitukai-o | utta |
| | | Someone-NOM | balcony-LOC | is | actress$_{[LOW]}$-GEN | servant$_{[HIGH]}$-ACC | shot |

For instance, Japanese and Korean native speakers prefer HA interpretations, attaching relative clauses to a higher NP, that is, *the servant* in (2). As head-final languages, the word order of the head noun and the relative clause in Japanese and Korean is opposite from English, meaning a lower NP precedes a higher NP (i.e., "actress" + "servant").

While interpretation preferences are often assessed through offline tasks that require judgments after reading entire sentences, psycholinguistic research has examined interpretation preferences in real-time processing to assess initial interpretations. For English, numerous studies have consistently reported a LA preference in both online (real-time) and offline language processing, indicating a stable parsing strategy [48,71]. In Korean, a HA preference persists across both the initial parsing and final interpretation stages, demonstrating stability throughout processing [59,61]. Japanese, on the other hand, is known to deviate from these stable patterns, exhibiting a shift in attachment preference between processing stages: speakers initially favor an LA interpretation during online processing, but ultimately resolve the ambiguity with an HA interpretation by the time of offline judgment [64]. A similar pattern has been reported for Spanish, where an initial LA preference often shifts to HA in final interpretations [48,51]. However, this early LA preference is not always robust. Another study in Spanish, particularly those manipulating linguistic factors such as animacy, have shown that HA preferences can also emerge during real-time processing [72]. Thus, early parsing strategies may vary based on both language-specific tendencies and structural or semantic cues [70]. Table 1 summarizes the general attachment preferences in four languages.

## Ambiguity in large language models

The performance of language models (LMs) on ambiguous sentences remains an underexplored area. Some studies have examined encoder models, primarily with English data. Recently, [73] reported that LMs' predictability, based on surprisal, aligns with human processing difficulty for a variety of syntactically complex English sentences, including attachment ambiguities. Wiki-LSTM and GPT-2 generally predicted the direction of human processing difficulty, but did not consistently predict its magnitude across sentence types. For sentences with attachment ambiguities, they compared processing difficulty between HA and LA interpretations at the word following the critical verb region. The models

**Table 1. Summary of previous studies testing relative clause attachment ambiguity.**

| Language | Offline interpretation | Online real-time processing |
|---|---|---|
| **English** | LA | LA |
| **Spanish** | HA | LA |
| **Japanese** | HA | LA |
| **Korean** | HA | HA |

 

accurately predicted where difficulty would arise, showing increased processing difficulty for HA interpretations but not for LA interpretations.

Davis (2022) [74] further analyzed attachment ambiguity by examining Long Short-Term Memory (LSTM) models and various transformer models (GPT-2 XL, BERT, and RoBERTa) in English and Spanish. Overall, these LLMs showed a preference for LA interpretations in English (LSTMs, GPT-2 XL, and BERT), though RoBERTa did not exhibit a clear preference. Recall that the native Spanish speakers have a different preference from English, favoring HA attachment. However, Spanish versions of GPT-2 and BERT tended to prefer LA attachment, similar to English models, while a subset of models, such as RuPERTa and the LSTMs, showed no preference. The preference for LA interpretation in Spanish may reflect the online processing behavior of Spanish speakers. However, Davis suggested an alternative explanation, proposing that the models might be capturing production trends as observed from the Spanish training corpus [75], rather than true comprehension tendencies.

While prior research has advanced our understanding of attachment ambiguities in both human cognition and language model performance, significant gaps remain. Most studies focus on high-resource languages such as English or Spanish, often neglecting typologically distinct languages like Korean and Japanese. Additionally, much of the research on LMs emphasizes their performance in isolated linguistic contexts without systematically exploring their adaptability to cross-linguistic variation in attachment preferences. Given the limited exploration of LMs' performance on resolving attachment ambiguities, we go one step further by employing a unified framework to assess both language-specific preferences and the broader question of whether LMs can meaningfully adapt to linguistic diversity.

## The present work

The present work uses attachment ambiguity as a testing ground to evaluate GPT's ability to capture language-specific interpretation preferences, focusing on GPT models' performance in three different languages: English, Korean, and Japanese. Specifically, we examine whether previous findings in the Spanish context simply reflected LA preferences based on English-language training—not language-independent or language-specific interpretation—genuinely capturing the online processing of Spanish interpretation preferences. Investigating GPT in Korean and Japanese allows us to further advance our understanding of its multilinguality. Recall that Japanese speakers initially prefer LA during online processing but shift toward HA in offline tasks, mirroring Spanish speakers when linguistic factors such as animacy or syntactic complexity are not involved. Conversely, Korean speakers consistently maintain an HA interpretation from the initial stages of parsing through to their final interpretations [59]. The divergence in initial parsing strategies between Japanese and Korean, despite their syntactic similarities, may reflect differences in parsing heuristics employed by speakers of these languages. For Korean, a persistent HA preference may arise from parsing mechanisms, such as strong sensitivity to syntactic cues, specifically predicate proximity [76], promoting attachment to structurally higher nouns even during the initial stages of sentence processing [59,61]. In contrast, Japanese parsing initially appears driven by a locality-based strategy, attaching the relative clause to the closest available noun phrase, resulting in early LA interpretations that later shift toward HA preferences [64]. We reason that if both Korean and Japanese exhibit LA preferences like in the English and (partial) Spanish cases, it indicates a limitation of LMs handling multiple languages and their bias toward English. However, if some findings of LA preference presented in Spanish captures online processing, we expect that only Japanese data should exhibit LA preference, while Korean data should not because Japanese only exhibited initial LA preferences [64].

## Materials and methods

For the main dataset, the experimental materials consisted of 210 items that were presented in 6 different versions for each language (1260 times). Some items were modified from stimuli used in [47] and [73]. We consistently used the pronoun 'I' as the main subject instead of person names because person names vary across languages, and we wanted to avoid any impact of familiarity of person names on interpretations. For Japanese, we consistently used *watasi*, which

is compatible with both male and female, instead of *boku* or *ore*, which are typically male-specific. To maximize control of sentences across three languages, we imposed several restrictions. First, the terms like *sister*, or *father* which describe relative relations were used only for the high attached nouns (NP1 in English, NP2 in Korean and Japanese) for natural-ness in English, like *the sister of the teacher*, not *the teacher of the sister*. If we use it in the high attached noun positions in English, NP2, it is more natural to use possessive like *my sister* as in *the teacher of my sister*, to refer to the sister of the main subject, *I* (without contexts). Otherwise, we used words for professions (e.g., *teacher, doctor*). Secondly, we ensured that both noun phrases (NPs) shared the same animacy to preserve ambiguity. In Japanese, there are two types of 'to be' verbs that differ based on animacy: *aru*, which indicates the existence of inanimate objects, and *iru*, which indicates the existence of animate beings. By maintaining the same animacy for both NPs, we avoided providing clues as to which subject the verb is modifying. The sample item for each language is exemplified in (3).

| (3) | a. | English: I observed the secretary of the clerk that was working. | | | | | |
|-----|-----|-----|-----|-----|-----|-----|-----|
| | b. | Korean: | | | | | |
| | | Na-nun | il-ha-ko | iss-nun | samwukwan-uy | pise-lul | cikhyepo-ass-ta. | |
| | | I-TOP | work-do-PROG | is-ADN | clerk-GEN | secretary-ACC | observe-PST-DCL | |
| | c. | Japanese: | | | | | |
| | | Watasi-wa | sigoto | | si-te iru | zimukan-no | hisyo-wo | mimamot-ta. |
| | | I-TOP | work | | do-PROG | clerk-GEN | secretary-ACC | observe-PST |

Previous studies have highlighted that LLMs are sensitive to question formatting, particularly in multiple-choice settings, where their responses can be influenced by the structure of the prompt and the order in which answer choices are presented [77–81]. To avoid these effects, we created three question templates (*A/B*, *Compare,* and *Repeat*) with two different orders of answer choices or question types, as presented in (4), following [82] associated with ambiguous scenarios regarding moral belief choices. For Question types A and C, we systematically varied the order in which the two noun phrases were presented: in one condition, NP1 appeared as the first option, while in the other condition, NP2 appeared first. In the *yes/no* format used for Question type B, we similarly manipulated the focus of the query by explicitly asking not only whether NP1 was the modified noun phrase but also whether NP2 was the modified entity. Therefore, 210 set of items were asked twice by manipulating the order of answers, thus each question type was asked 420 times, making a total 1260 items. (4) presents the set of sample prompt templates in English.

| (4) | Sample sentence: "I observed the secretary of the clerk that was working." |
|-----|-----|
| | a. Question type A: A/B |
| | Question: In the following sentence, which one was working? Respond solely with 'A' or 'B'. |
| | A: the secretary |
| | B: the clerk |
| | b. Question type B: Compare |
| | Question: In the following sentence, is the secretary the one that was working? Respond solely with 'yes' or 'no'. |
| | c. Question type C: Repeat |
| | Question: Which of the following is the one that was working? Respond solely by repeating one of the following options exactly. |
| | A: the secretary that was working |
| | B: the clerk that was working |

Question type 1 asks to select a modified NP and Question type 2 directly asks whether NP1 or NP2 is the one performing an action or in the state described by relative clauses by answering yes/no. Question type 3 asks to repeat the full phrases, the modified NPs with relative clauses. We conducted this experiment through the OpenAI application programming interface (API), an open access online service. We also provided instructions in each language to force GPT models to process language specifically because previous studies found that GPT is better at analyzing tasks with English prompts [35]. In all responses, we additionally asked GPT models to state the reason why it selected a specific answer; this aspect is discussed in the Discussion section.

We examined five GPT models—GPT-3.5-turbo, GPT-4-turbo, GPT-4o, o1-mini, and o3-mini. GPT-3.5-turbo serves as a foundational model commonly used for benchmarking due to its computational efficiency and robust general performance across various language-processing tasks [83]. GPT-4-turbo, is an advanced successor, notable for improved reasoning, enhanced contextual understanding, and greater accuracy, especially in complex linguistic contexts [84]. GPT-4o, representing the newer generation, incorporates multimodal processing and improved multilingual functionality, specifically aiming to mitigate language-specific biases and enhance adaptability to diverse linguistic contexts [85]. In addition to these mainstream models, we included two recently released lightweight variants—o1-mini and o3-mini—developed to provide faster and more cost-efficient alternatives while maintaining competitive performance in reasoning tasks. The o1-mini model, released in September 2024, prioritizes enhanced reasoning by incorporating a chain-of-thought approach, facilitating better handling of complex, logic-intensive tasks. The o3-mini model, introduced in January 2025, is streamlined to deliver prompt, accurate responses tailored for instructional and interactive real-time applications. Their inclusion provides novel insights, as performance on complex linguistic phenomena like structural ambiguity remains underexplored in these recent models. Furthermore, evaluating o1-mini and o3-mini enables us to explore how variations in training techniques and model scale affect syntactic ambiguity resolution and multilingual processing, as detailed in the Discussion section.

## Analysis and results

To analyze whether one interpretation (A or B) was significantly preferred over the other, we initially ran separate logistic mixed-effects regression models for each prompt template using the *glmer* function from the *lme4* packages [86] in R ([87]). The preference of interpretation (binary: LA or HA) was set as the dependent variable, with the probability of choosing interpretations. To examine interpretation preferences across all prompt templates, we first ran a combined logistic mixed-effects regression model incorporating all templates into the analysis. This model included a random intercept to control for variability across individual prompt templates, alongside random intercepts for item and the order of answer choices (Question type A, C) or the noun being questioned (Question type B). This allowed us to control for variability across individual prompt templates, the order of options, and the items presented. Next, we conducted a sub-analysis to determine whether one interpretation was significantly preferred over the other for each prompt template individually. For this, we ran separate logistic mixed-effects regression models, including two random intercepts to account for variability associated with item and order of choices. The results of the models are presented in Table 2.

The proportion of LA preferences by model for each language is plotted in Fig 1. *P* values were determined using a Satterthwaite approximation through the computation of the *LmerTest* function [88].

We report pooled results across all question types for each language, followed by separate analyses for each type. In English, all GPT models from GPT-3.5-turbo to o3-mini consistently showed a general preference for LA interpretations across question types, aligning with native English speakers' tendencies to agree with the lower noun. These trends were validated by statistical significance in the analysis. In the subset analysis, LA preferences were significant across question types, with one exception. Question type B in GPT −4-turbo exhibited the same trend, favoring LA interpretations, but did not reach statistical significance ($z = -0.83$, $p = 0.40$).

One plausible explanation for this lack of statistical significance is variability arising from the random effect of question type. Specifically, GPT-4-turbo exhibited a strong bias related to which noun phrase was queried: when explicitly asked

**Table 2. Mean proportions of LA interpretations.**

| Model | Question type | Language | | |
|---|---|---|---|---|
| | | English | Korean | Japanese |
| o3-mini | All types | 90.48% (1140/1260)** | 78.01% (983/1260)** | 83.09% (1047/1260)** |
| | Question type A | 92.38% (388/420)** | 78.33% (329/420)** | 80.71% (339/420)** |
| | Question type B | 92.38% (388/420)** | 68.80% (289/420)** | 75.95% (319/420)** |
| | Question type C | 86.66% (364/420)** | 86.90% (365/420)** | 92.62% (389/420)** |
| o1-mini | All types | 93.49% (1178/1260)** | 58.49% (737/1260) | 69.20% (872/1260) ** |
| | Question type A | 90.95% (382/420)** | 56.42% (237/420)** | 61.42% (258/420)** |
| | Question type B | 91.66% (385/420)** | 49.76% (209/420) | 61.19% (257/420)** |
| | Question type C | 97.86% (411/420)** | 69.28% (291/420)** | 85% (357/420)** |
| GPT-4o | All types | 73.17% (922/1260)** | 64.60% (814/1260)** | 45.79% (577/1260)** |
| | Question type A | 72. 85% (306/420)** | 67. 61% (284/420)** | 42.61% (179/420)** |
| | Question type B | 70.81% (296/420)** | 58.33% (245/420)** | 40.71% (171/420)* |
| | Question type C | 76.19% (320/420)** | 67.85% (285/420)** | 54.04% (227/420) |
| GPT-4-turbo | All types | 68.88% (868/1260)** | 75.15% (947/1260) ** | 54.20% (683/1260) |
| | Question type A | 72.38% (304/420)** | 74.76% (314/420)** | 54.52% (229/420) |
| | Question type B | 63.57% (267/420) | 72.85% (306/420)** | 50.95% (214/420) |
| | Question type C | 70.71% (297/420)** | 77.85% (327/420)** | 57.14% (240/420) |
| GPT-3.5-turbo | All types | 73.73% (929/1260)** | 72.14% (909/1260)* | 61.58% (776/1260) |
| | Question type A | 76.90% (323/420)** | 75.95% (319/420) | 59.52% (250/420) |
| | Question type B | 62.38% (262/420)** | 60.23% (253/420) | 49.52% (208/420) |
| | Question type C | 81.90% (344/420)** | 80.23% (337/420)** | 75.71% (318/420)** |

Asterisks (**) indicate significant differences and (*) indicate marginal differences between LA and HA interpretations ('**' indicates $p < 0.05$ and '*' indicates a $p$-value between 0.05 and 0.10).

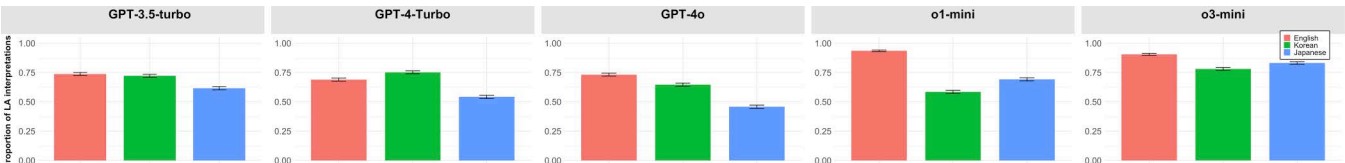

**Fig 1. Mean proportion of the LA interpretation across three languages: English, Korean, and Japanese.** Error bars indicate the standard error of the mean.

if NP1 was modified, the model selected NP2 in over 90% of cases, whereas when NP2 was the explicit query, NP1 was preferred in over 65% of responses. This pattern was further supported by statistical analysis, which revealed a significant effect when the random intercept for the noun being questioned was excluded from the model ($z = -5.49$, $p < 0.01$). This pattern suggests that GPT-4-turbo's attachment interpretations were significantly influenced by the particular noun phrase queried. One possible account for the increased variability and lack of statistical significance observed in the yes/no question format (Question Type B) may lie in the inherent property of this prompt structure. Unlike Question Types A and C, which present contrasting alternatives and require the model to resolve ambiguity by selecting between interpretations, yes/no prompts require the model to evaluate a single interpretation in isolation. This lack of direct comparison may result in weaker commitment to a specific interpretation, producing more variable responses, leading to reduced robustness in

attachment preferences. Prior studies have shown that LLMs are sensitive to subtle variations in phrasing or prompt structure in yes/no tasks, which can further exacerbate inconsistencies and limit interpretive clarity [81,89].

For Korean, all GPT versions similarly showed a preference for LA interpretations, with over 60% of responses favoring this interpretation—mirroring the English data. Although the overall LA interpretation rate of o1-mini did not reach 60%, it was close—reaching 58.49%—and still showed a numerical trend toward LA preference. Statistical analysis confirmed a significant preference for LA over HA in Korean, with marginal significance in GPT-3.5-turbo ($z = -1.63$, $p = 0.10$). The sub-analysis for each question type reached statistical significance for GPT-4o and GPT-4-turbo. Although question types A and B in GPT 3.5 did not achieve statistical significance, LA preferences were still evident, with over 75% favoring LA interpretations for question type A and 60% for question type B. The interpretation for Question Type B in o1-mini was also not statistically significant, which likely contributed to the lack of significance in the overall results. However, when the random intercept for question type was excluded from the model, the overall interpretation became significant ($z = 3.03$, $p = 0.002$), mirroring the pattern observed with Question type B in GPT-4-turbo.

In Japanese, GPT models from GPT-3.5-turbo to GPT 4o generally showed no strong preference for either LA or HA interpretations, with overall differences remaining insignificant. However, in the sub-analyses, only question type C in GPT 3.5 displayed a significant preference for LA (over 75%), reflecting trends similar to those observed in English and Korean. Interestingly, GPT-4o even showed an overall significant preference for HA interpretations, which is the opposite of the trends in English and Korean. Additionally, with the release of newer GPT models, we observed a gradual decrease in the proportion of LA interpretations, particularly evident in GPT-4o. This shift clearly supports that GPT-4o's intended focus on enhanced multilingual capability has indeed been realized, as it begins to reflect human-like interpretation patterns in Japanese. However, the more recent smaller-scale models, o1-mini and o3-mini, exhibited statistically significant preferences for LA interpretations, mirroring the patterns observed in English.

## Discussion

### Cross-linguistic model comparisons

A fundamental question of the current study was to evaluate whether GPT models can be effectively applied across languages or if language-specific technologies are needed for non-English languages. To address this, we examined GPT models' cross-linguistic performance in a linguistic domain that requires nuanced understanding of human language processing. Our findings revealed that GPT models display human-like behavior in English, showing a preference for LA interpretations. Despite the cross-linguistic differences evident among Korean native speakers, we observed that GPT models mirrored English native speaker preferences, favoring LA interpretations—a tendency not shared by Korean speakers.

This result raises important questions about whether LMs reflect online processing or if their responses are influenced mainly by training data in English. As discussed in the introduction, we anticipated that if the LA preference in Spanish, noted by Davis (2022) [74], genuinely reflected initial parsing, only Japanese data would demonstrate an LA preference, while Korean data would not, given that Korean speakers exhibit HA preferences in both online and offline settings. However, our observation of consistent LA preferences in Korean suggests that this tendency in Spanish ambiguous sentences may not reflect online processing. Instead, the LA preferences in Korean suggest that GPT's interpretation of Korean is heavily influenced by its English-based training, possibly affecting Spanish as well. To validate this conclusion further, future research could directly compare Spanish data with Korean or other languages with similar HA patterns to Korean. Additionally, as prior studies and the current one have evaluated distinct architectures—encoder and decoder models—further research would benefit from a comprehensive examination of both. It is also noteworthy that our study evaluated more recent models compared to Davis' (2022) work [74] with the GPT-2 XL model.

Interestingly, the Japanese data did not show a strong preference for either interpretation GPT 3.5-turbo and 4-turbo. This could be due to several factors. One possibility is resource availability. According to Common Crawl corpus and its

snapshot—a large-scale, non-curated dataset of multilingual webpages that is used to pre-train models like GPT-3—Japanese is considered a higher-resource language compared to Korean [34,35,90]. As Japanese falls into a higher-resource category, the availability of more data likely helps models suppress a direct reliance on English-based training, in contrast to Korean, which may be more susceptible to mirroring of English speakers' interpretation preferences due to resource limitations. The gradual increase in HA interpretations, from 38.42% to 54.21% as GPT models advance with evidence of its significance in two question types in the latest model 4o, may indicate a progressive development of human-like interpretive tendencies, possibly influenced by the availability of relatively high resources. This shift clearly demonstrates that GPT-4o's intended enhancements in multilingual capability have been realized, enabling it to more closely reflect native Japanese interpretation patterns and highlighting the benefits associated with high resources in this model. Another possible explanation is that certain Japanese sentences could have alternative interpretations. For example, nouns indicating relational roles, like *son* in phrases such as *the son of the architect*, could create ambiguities, where *architect* is interpreted as an appositive describing the son's profession. These potentially confounding sentences comprised 35.23% of the total items, or 74 out of 210 items in our stimuli. To address this, we conducted an exploratory analysis on the Japanese data, excluding these potentially confounding sentences. The results confirmed the initial trend, with no significant preference for either interpretation across all GPT models (all $ps > 0.36$). Therefore, the lack of strong preferences in Japanese cannot be entirely explained by these sentence types.

This absence of clear attachment preferences in GPT-3.5-turbo and GPT-4-turbo for Japanese may reflect the influence of relatively higher resource availability for Japanese compared to Korean during training. This broader exposure may have mitigated the tendency to generalize syntactic patterns learned from English, leading to more balanced behavior. In this context, GPT-4o, which was explicitly trained to improve cross-linguistic generalization and reduce English-language bias, appears better suited for capturing language-specific patterns of attachment ambiguity in relatively resource-rich languages like Japanese.

The robust LA preferences observed in o1-mini and o3-mini for Japanese as well as Korean—mirroring patterns found in English—further suggest the importance of training data scale and techniques in shaping interpretive behavior. These smaller-scale models, optimized for speed and cost-efficiency, contain significantly fewer parameters and are likely exposed to a narrower range of non-English linguistic input during training. Consequently, they appear to rely more heavily on generalized heuristics learned from high-resource languages like English, consistently favoring LA interpretations across typologically diverse languages. This behavior reflects a trade-off, where deep contextual reasoning is sacrificed in favor of faster inference, often prioritizing surface-level generalizations and high-probability completions. These findings align with prior research showing that model performance improves with increased size and data exposure [91,92], suggesting that smaller models may be less equipped to capture language-specific syntactic nuances in multilingual contexts.

In terms of tuning techniques, the reduced multilingual capability of o1-mini and o3-mini can be attributed to the techniques used for model alignment. While GPT-3.5-turbo, GPT-4-turbo, and GPT-4o were trained using Reinforcement Learning from Human Feedback (RLHF), as well as instruction-tuning, o1-mini and o3-mini are mainly instruction-tuned [93]. RLHF allows models to be further refined using human feedback, enhancing their ability to produce responses that align with human preferences—particularly in cases of ambiguity or underspecified input. This additional layer of alignment improves flexible reasoning, contextual coherence, and interpretive sensitivity. In the absence of RLHF, o1-mini and o3-mini may be more prone to rely more heavily on surface-level heuristics or high-frequency patterns seen during pretraining, limiting their ability to capture subtle syntactic and semantic distinctions across languages and reinforcing English-based interpretations.

## Model reasoning and future directions

As noted earlier, we asked GPT models to provide the reasoning behind each answer choice, enabling us to examine how GPT approaches the resolution of attachment ambiguity. For English, GPT predominantly selects the LA interpretation

because NP2 (i.e., the low attachment) is positioned nearer to the relative clause, with the general explanation that "relative clauses typically modify the nearest preceding noun." This reasoning suggests that GPT identifies a pattern of LA preferences based on the linear distance between the relative clause and a noun, as learned from English data. This reasoning also applies to Korean and Japanese interpretations, where GPT selects the LA interpretation by identifying the closest noun. Despite the differences in word order from English, the relative clauses in both Korean and Japanese are also positioned closer to the noun associated with low attachment (i.e., NP1). Notably, even when GPT occasionally chooses high attachment nouns, it still provides the same explanation about the noun's proximity to the relative clause, which is clearly incorrect. While Japanese interpretations may appear less influenced by English data in our binary task, GPT's reasoning process is not entirely language-independent.

Another possibility is that GPT's provided rationales may not reflect the actual decision-making criteria employed during ambiguity resolution, but rather constitute post-hoc rationalizations disconnected from its underlying interpretive processes. In other words, GPT might simply generate justifications based on generalized heuristics (such as predicate proximity-based attachment), independent of whether these heuristics genuinely guided its specific attachment choices. This possibility suggests that GPT's explanations might not offer reliable insights into its decision reasoning, but instead reflect plausible, learned patterns drawn from training data regarding typical human-like linguistic reasoning. Such a scenario aligns with recent concerns expressed by [94], who caution against conflating LLMs' apparent linguistic competence with genuine reasoning capacities. Thus, while GPT's explicit reasoning offers some insight into its linguistic behavior, it may not directly reflect the underlying processes guiding its decisions. Future research could further investigate whether GPT's provided rationales are genuinely connected to its interpretive decisions or merely represent surface-level justifications, by employing more sophisticated experimental tasks and analytical measures.

Our findings suggest that GPT models' processing aligns with the Late Closure strategy [71], one of the well-known psycholinguistic theories for resolving attachment ambiguity. According to this principle, if grammatically permissible, incoming words are likely attached to the most recent phrase or clause being processed, rather than an earlier one in the sentence. Thus, when encountering structural ambiguity, readers are inclined to connect the relative clause to the nearest noun phrase (NP) rather than one farther back. While this processing strategy initially aimed to explain attachment preferences universally, it encountered limitations as studies revealed cross-linguistic differences. For GPT models, it seems that the Late Closure strategy is overgeneralized across languages, without incorporating language-specific patterns. Even GPT-4, one of the latest models, did not fully adopt language-specific interpretations, indicating that it may not be entirely multilingual.

Although multilingual data enables GPT to handle inputs and produce responses across different languages, our findings highlight the gap between GPT's current capabilities and the nuanced adaptability required for genuine conversational sophistication. Addressing these limitations is critical for advancing GPT's multilingual functionality and its potential to achieve human-like communication skills. There is a clear need within the community for a comprehensive, public, and independent evaluation of GPT across diverse non-English languages and NLP tasks to inform future research and applications effectively. We propose several directions for future research. First, as our current study mainly evaluates GPT, subsequent studies could extend this analysis to include other recent multilingual models, such as LLaMa, BLOOM, or Claude. In fact, we conducted an exploratory evaluation using Anthropic's Claude 3.5 Sonnet, which revealed a similar trend across all three languages, each showing a consistent preference for LA interpretations exceeding 60%. As these findings extend beyond the primary focus of the present study, they are not included in the main manuscript but are provided in the Supporting Information for reference. This observed convergence across model families suggests that the LA preference may reflect a broader characteristic of LLMs, rather than being specific to the GPT architecture. Future research could systematically investigate whether this pattern holds across a wider range of LMM architectures under varied prompting conditions or other measurements to assess the extent to which current language models possess genuine multilingual capabilities.

Another research direction is conducting more in-depth cross-linguistic comparisons based on language families. For example, [95] recently highlighted the performance limitations of model editing techniques under a cross-lingual model editing paradigm, particularly when comparing languages from different language families, such as *Latin* and *Indic*. Building on this, future research could encompass comparisons with languages both within the same family as English, like *French* or *Dutch*, and those from distinctly different language families. Additionally, examining corpus data or human production tasks in Korean and Japanese could help clarify whether LA preferences observed in GPT align with production trends or comprehension tendencies in these languages. Such analyses would ensure a deeper understanding of interpretation trends and their alignment with human linguistic behavior. These broader approaches would enable a more comprehensive understanding of the impact of typological factors on model performance across languages.

## Conclusion

This study underscores the limitations of GPT's multilingual capabilities, particularly in resolving attachment ambiguities in non-English languages, where it tends to overgeneralize English-based processing strategies. While GPT can respond across languages, it lacks the language-specific nuance observed in human processing, as evidenced by its consistent LA preferences, even in typologically distinct languages like Korean and no preferences in Japanese. These findings suggest that language-specific technologies may be essential for building truly multilingual models, equipping them to reflect the structural nuances of each language for more accurate applications.

## Supporting information

**S1 Appendix. Experimental items.**
(XLSX)

**S2 Appendix. o1-mini dataset.**
(CSV)

**S3 Appendix. o3-mini dataset.**
(CSV)

**S4 Appendix. GPT-4o dataset.**
(CSV)

**S5 Appendix. GPT-4-turbo dataset.**
(CSV)

**S6 Appendix. GPT-3.5-turbo dataset.**
(CSV)

**S7 Appendix. Claude 3.7 Sonnet dataset.**
(CSV)

**S1 Table. The proportion of LA preferences by Claude 3.7 Sonnet.**
(PDF)

## Author contributions

**Conceptualization:** Myung Hye Yoo, Joungmin Kim, Sanghoun Song.

**Data curation:** Myung Hye Yoo, Joungmin Kim, Sanghoun Song.

**Funding acquisition:** Sanghoun Song.

**Supervision:** Sanghoun Song.

**Writing – original draft:** Myung Hye Yoo.

**Writing – review & editing:** Joungmin Kim, Sanghoun Song.

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
