## [Decision Letter · Decision Letter 0]

Dear Dr. Song,

Thank you for submitting your manuscript to PLOS ONE. After careful consideration, we feel that it has merit but does not fully meet PLOS ONE’s publication criteria as it currently stands. Therefore, we invite you to submit a revised version of the manuscript that addresses the points raised during the review process.

We look forward to receiving your revised manuscript.

Kind regards,

Montserrat Comesaña Vila

Academic Editor

PLOS ONE

**Journal Requirements:**

1. When submitting your revision, we need you to address these additional requirements. Please ensure that your manuscript meets PLOS ONE's style requirements, including those for file naming. The PLOS ONE style templates can be found at https://journals.plos.org/plosone/s/file?id=wjVg/PLOSOne_formatting_sample_main_body.pdf and https://journals.plos.org/plosone/s/file?id=ba62/PLOSOne_formatting_sample_title_authors_affiliations.pdf 2. Thank you for stating in your Funding Statement: This work was supported by the NRF under the project BK21 FOUR(4299990414427). Please provide an amended statement that declares *all* the funding or sources of support (whether external or internal to your organization) received during this study, as detailed online in our guide for authors at http://journals.plos.org/plosone/s/submit-now.  Please also include the statement “There was no additional external funding received for this study.” in your updated Funding Statement. Please include your amended Funding Statement within your cover letter. We will change the online submission form on your behalf.

**Additional Editor Comments:**

Dear authors:

I'm writing about your submission to Plos One. The comments on your manuscript from reviewers whose expertise falls within the area of your research are now available.

As you can see from the attached, while the reviewers comment that your study has the potential to contribute significantly to the literature, they have raised several theoretical and methodological issues with it. Reviewer 1 is more enthusiastic than Reviewer 2. I also read your manuscript carefully. I agree with both reviewers that the manuscript is interesting. I also share the main concern of Reviewer 2 about the fact that as you have focused on a single version of GPT, you need to widen the scope of the manuscript.

You can see the comments of the Reviewers and consider them in order to improve the manuscript. You can also see my specific comments below:

Line 76, there is a typo: GPT mdoels´ instead of GPT models

Lines 100 to 105. You may want to also include European and Brazilian Portuguese:

Soares, A. P., Fraga, I., Comesaña, M., & Piñeiro, A. (2010). El papel de la animacidad en la desambiguación de oraciones de relativo en portugués europeo: evidencia en tareas de producción y comprensión. Psichotema, 22(4), 691-696.

Soares, A. P., Oliveira, H., Ferreira, M., Comesaña, M., Macedo, F., Ferré, P., Acuña-Fariña, J. C., Hernández Cabrera, J., & Fraga, I. (2019). Lexico-syntactic interactions during the processing of temporally ambiguous L2 relative clauses: An eye-tracking study with intermediate and advanced Portuguese-English bilinguals. Plos One, 14(5): e0216779. DOI: 10.1371/journal.pone.0216779

Maia, M., Fernández, E. M., Costa, A., & do Carmo Lourenço-Gomes, M. (2007). Early and late preferences in relative clause attachment in Portuguese and Spanish. Journal of Portuguese Linguistics, 6(1).

Lines 122 to 125. There are more studies that should be included here to soften this statement. See for instance

Acuña-Fariña, C., Fraga, I., García-Orza, J., & Piñeiro, A. (2009). Animacy in the adjunction of Spanish RCs to complex NPs. European Journal of Cognitive Psychology, 21(8), 1137-1165.

In this study, the authors showed a clear HA strategy that is reversed as a function of animacy. I mean, there are several studies using online measures in which the HA preference has been observed.

Lines 169 to 172. You should soften this statement considering my previous comments. Also, a brief explanation of why Korean show a different trend than Japanese would be appreciated.

Lines 185 to 187. If the terms were used for the low attached nouns, how sister can be in NP1? I lost the point here, please clarify it.

Line 227. There is another typo. An extra dot after "one"

Best wishes,

Montserrat Comesaña

Reviewers' comments:

Reviewer's Responses to Questions

**Comments to the Author**

1. Is the manuscript technically sound, and do the data support the conclusions?

Reviewer #1: Yes

Reviewer #2: Partly

2. Has the statistical analysis been performed appropriately and rigorously?

Reviewer #1: Yes

Reviewer #2: N/A

3. Have the authors made all data underlying the findings in their manuscript fully available?

Reviewer #1: Yes

Reviewer #2: Yes

4. Is the manuscript presented in an intelligible fashion and written in standard English?

Reviewer #1: Yes

Reviewer #2: Yes

**Reviewer #1:**  This study takes into account the attachment preferences of English, Japanese, and Korean speakers and compares them with those of three versions of the GPT model in the same languages. The goal is to determine whether GPT models exhibit syntactic behavior similar to that of native speakers or if they instead show a bias toward English, the language in which they have received the most training. The findings support the latter hypothesis, as GPT’s attachment preference in Korean diverges from that of native speakers, and no clear preference is observed in Japanese.

I was able to follow the manuscript easily and the content is interesting, although I think there are some clarifications that could be made to make it easier to follow. I therefore have only a few minor comments to make in this review, and I would be grateful if they could be taken into account:

1.- Page 5, lines 76 to 83. I think the whole paragraph is out of place, as it belongs more in the discussion or conclusions than in an introduction. The easiest thing to do would be to delete it.

2.- On page 12, Materials and methods, it says that three different versions of the GPT were used. It would be good to give a reason for this, or at least what is different/interesting about the three versions used.

3.- While the results for English are very clear and consistent, question type B in GPT -4 turbo was not significant, while question type B for GPT -3.5 turbo, with a lower percentage of LA responses, was. Any idea what this might be due to?

4.- Pages 18-19, lines 338-350. The GPT models were asked to give an explanation for each answer choice. According to the text, even when the preferred option was HA, the models gave the explanation that 'relative clauses typically modify the nearest preceding noun'. This is interpreted to mean that their reasoning process is not completely language independent. However, I see another possible explanation here: GPT may not be answering on the basis of the criteria it actually used to make the decision, because it is unaware of them. In other words, its answer and the justification could be independent of each other, since it seems to generate explanations uniformly based on a general knowledge of how these ambiguities are usually resolved, rather than on the specific case of the item presented. If this is the case, this part of the discussion would not provide much information about its actual reasoning. I would like to hear about this possibility.

Typos:

Page 3, line 26: mdoels’ -> models’

Page 5, line 85: The abbreviation RC here is the first time it appears, and it is not used again in the whole article. So, it is better to replace it with the full wording: relative clause.

Page 6, line 107: HA preferences). -> remove )

Page 8, line 142: Indicate what the acronym LSTM stands for.

Page 10, line 169: multiliguality -> multilinguality.

Page 11, lines 205-207: rephrase the sentence.

Page 12, line: 227: remove the full stop at the end.

Page 13, line 239: different reference format!

Figure 1. Please, increase figure quality.

**Reviewer #2:**  Review of ms “Multilingual Capabilities of GPT: A Study of Structural Ambiguity”

The topic of structural ambiguity in large language models is timely and interesting. However, the current version is too focused on GPT, which is continuously evolving, meaning the present patterns may or may not replicate in 2025 versions or newer. As acknowledged by the authors, the study would need to be extended to other large language models, which may have been trained differently.

Perhaps, adding other current models (or even different versions of the same model, such as GPT-4.0 with o1 or o3) could help the manuscript cross the publication boundary at PLOS One. However, this would require rethinking the overall approach.

**Do you want your identity to be public for this peer review?** For information about this choice, including consent withdrawal, please see our Privacy Policy

Reviewer #1: No

Reviewer #2: No

---

## [Author Response · Author response to Decision Letter 1]

1 May 2025

We have attached the file additionally.

---

## [Decision Letter · Decision Letter 1]

Multilingual Capabilities of GPT: A Study of Structural Ambiguity

PONE-D-24-56759R1

Dear Dr. Song,

We’re pleased to inform you that your manuscript has been judged scientifically suitable for publication and will be formally accepted for publication once it meets all outstanding technical requirements.

Kind regards,

Montserrat Comesaña Vila

Academic Editor

PLOS ONE

Reviewer #1: All comments have been addressed

2. Is the manuscript technically sound, and do the data support the conclusions?

Reviewer #1: Yes

3. Has the statistical analysis been performed appropriately and rigorously?

Reviewer #1: Yes

4. Have the authors made all data underlying the findings in their manuscript fully available?

Reviewer #1: Yes

5. Is the manuscript presented in an intelligible fashion and written in standard English?

Reviewer #1: Yes

Reviewer #1: All my main concerns have been perfectly addressed in this new version of the manuscript. In addition, new data and theoretical considerations are presented. I have not detected any new typos in this version. Therefore, I have no further comments to make.

**Do you want your identity to be public for this peer review?** For information about this choice, including consent withdrawal, please see our Privacy Policy

Reviewer #1: No

---

## [Editor Report · Acceptance letter]

PONE-D-24-56759R1

PLOS ONE

Dear Dr. Song,

I'm pleased to inform you that your manuscript has been deemed suitable for publication in PLOS ONE. Congratulations! Your manuscript is now being handed over to our production team.

Kind regards,

on behalf of

Dr. Montserrat Comesaña Vila

Academic Editor

PLOS ONE